# Therapeutic Drug Monitoring-Guided Linezolid Therapy for the Treatment of Multiple Staphylococcal Brain Abscesses in a 3-Month-Old Infant

**DOI:** 10.3390/pathogens14010004

**Published:** 2024-12-27

**Authors:** Anna Cascone, Maia De Luca, Raffaele Simeoli, Bianca Maria Goffredo, Laura Cursi, Costanza Tripiciano, Lorenza Romani, Stefania Mercadante, Martina Di Giuseppe, Francesca Ippolita Calo Carducci, Davide Luglietto, Paola Bernaschi, Laura Lancella

**Affiliations:** 1Residency School of Pediatrics, Department of Systems Medicine, University of Rome “Tor Vergata”, 00133 Rome, Italy; 2Infectious Disease Unit, Bambino Gesù Children’s Hospital, IRCCS, 00165 Rome, Italy; maia.deluca@opbg.net (M.D.L.); laura.cursi@opbg.net (L.C.); costanza.tripiciano@opbg.net (C.T.); lorenza.romani@opbg.net (L.R.); stefania.mercadante@opbg.net (S.M.); martina.digiuseppe@opbg.net (M.D.G.); fippolita.calo@opbg.net (F.I.C.C.); laura.lancella@opbg.net (L.L.); 3Division of Metabolic Diseases and Drug Biology, Bambino Gesù Children’s Hospital, IRCCS, 00165 Rome, Italy; raffaele.simeoli@opbg.net (R.S.); biancamaria.goffredo@opbg.net (B.M.G.); 4Neurosurgery Unit, Bambino Gesù Children’s Hospital, IRCCS, 00165 Rome, Italy; davide.luglietto@opbg.net; 5Microbiology and Diagnostic Immunology Unit, Bambino Gesù Children’s Hospital, IRCCS, 00165 Rome, Italy; paola.bernaschi@opbg.net

**Keywords:** brain abscess, methicillin-sensitive *Staphylococcus aureus* (MSSA), therapeutic drug monitoring (TDM), linezolid, multidisciplinary approach

## Abstract

Brain abscesses are invasive infections of the central nervous system with a high level of treatment complexity especially in pediatric patients. Here, we describe a 3-month-old infant with multiple brain abscesses caused by methicillin-susceptible *Staphylococcus aureus* (MSSA). The patient was initially treated with empirical antibiotics (ceftriaxone, metronidazole, vancomycin). Upon MSSA identification, therapy was optimized by switching vancomycin to linezolid to improve tissue penetration. Therapeutic drug monitoring (TDM) was performed to check linezolid levels in the plasma and pus of the abscess, confirming drug penetration into brain tissue. A two-stage surgical drainage approach, consisting of repeated pus aspiration through an intracystic catheter, was then performed to achieve a significant reduction in abscess size. After nine weeks of antibiotic therapy, the patient was discharged in good clinical condition. This case highlights the role of linezolid for the treatment of complicated CNS infections and the importance of a multidisciplinary approach, combining TDM-based antibiotic therapy with timely and eventually repeated surgery, in order to effectively treat brain abscesses.

## 1. Introduction

Brain abscesses are invasive infections of the central nervous system that cause significant management challenges. The incidence in the general population is estimated between 0.4 and 1.3 per 100,000 inhabitants, which translate to roughly 6700 cases per year in Europe. While data on the incidence in the pediatric population are sparse, it is estimated that approximately 25% of brain abscesses occur in children [1].

The formation of an abscess can occur in a subacute manner and typically takes about 2 weeks, beginning with an initial stage of cerebritis, characterized by localized infection without a capsule or pus, and further progressing to the development of an abscess made of an outer capsule and purulent material inside [2].

The primary causative agents for brain abscesses in the pediatric population are *Streptococcus* spp. (36% of cases), followed by *Staphylococcus* spp. (18% of cases) [3,4]. This can be readily explained by considering that the intracranial spread of these pathogens occurs in half of the cases (50%) through contiguity, originating from an infectious focus in the maxillofacial region such as acute otitis media, sinusitis, or oral cavity infections, or through cranial discontinuities (post-traumatic or post-surgical). In 30–40% of cases, they disseminate via the bloodstream from a distant focus of infection [1]. Less frequently, the agents responsible are Gram-negative enteric bacteria and anaerobic bacteria [4].

Predisposing conditions include cranial trauma, cyanotic congenital heart disease, and chronic ear infections, but in recent decades, dental infections and immunocompromised status have also emerged as important risk factors [5]. Prematurity is commonly recognized as a risk factor for the development of severe infections. In fact, the immune system immaturity combined with the medical interventions required to support a premature neonate may lead to an increased risk of infections during the first year of life [6].

The clinical triad of symptoms suggesting a brain abscess is a fever, headache, and neurological deficit. However, in neonates, bulging fontanelle and/or an increased head circumference may also occur [7].

Here, we present the case of an infant with multiple brain abscesses caused by methicillin-susceptible *Staphylococcus aureus* (MSSA) that required a combined medical–surgical approach to be successfully treated.

## 2. Materials and Methods

### 2.1. Therapeutic Drug Monitoring

Linezolid concentrations in both pus and plasma samples were measured at the Laboratory of Metabolic Diseases and Drug Biology of Bambino Gesù Children’s Hospital, IRCCS, in Rome.

The ultra-high-performance liquid chromatography (UHPLC) apparatus used for the determination of linezolid levels consisted of an Agilent 1290 Infinity II system equipped with a quaternary pump, a degassing line, a thermostated autosampler, a column oven, and a 10 μL cell DAD (Diode Array Detector) (Agilent Technologies, Deutschland GmbH, Waldbronn, Germany). The analysis of linezolid concentrations was carried out by using a CE/IVD-validated HPLC kit (antibiotics in serum/plasma) provided by Chromsystems (Chromsystems Instruments & Chemicals GmbH, Gräfelfing, Germany). This analytical kit included calibrators and both Low- and High-Quality Controls (QCs) at fixed concentrations of 7.96 and 22.3 µg/mL, respectively. Each batch of patients’ samples included both calibrators and QCs prepared according to the manufacturer’s instructions. Data were acquired and processed by using OpenLAB Workstation (Agilent Technologies, Waldbronn, Germany). Samples with drugs’ concentrations above the higher calibration point were diluted and re-analyzed once again.

### 2.2. Informed Consent Form

Informed consent was obtained from the patient’s parents for the publication of this paper.

## 3. Detailed Case Description

We present the case of a 3-month-old male infant, born from an uncomplicated pregnancy at 35 + 4 weeks due to the premature rupture of membranes, via vaginal delivery. Vagino-rectal swabs and TORCH tests were negative; the Apgar score was 9–10. At birth, he was hospitalized for 10 days due to respiratory distress that did not require intubation, and indirect hyperbilirubinemia, which necessitated phototherapy. During the hospitalization, a peripheral venous access was placed for the administration of intravenous therapy; moreover, a brain ultrasound was performed, which was unremarkable except for the presence of a subcentimetric cyst at the left choroid plexus.

At 16 days of life, the patient was admitted due to *Staphylococcus aureus* bacteremia and cellulitis of the right hand, at the site of a previous peripheral venous access, and was treated for ten days.

At the age of 3 months, he was admitted to the Emergency Department of our hospital due to macrocrania and bulging fontanelle. His head circumference increased from the 12th to the 97th percentile within a month, while the other auxological parameters progressed regularly. He was afebrile, showing no neurological signs or other clinical symptoms. Blood tests revealed leucocytosis and elevated inflammatory markers (C-reactive protein: 3.8 mg/dL, n.v. <0.5 mg/dL). The brain MRI scan revealed the presence of multiple rounded formations in the right hemisphere (Figure 1a), characterized by the alteration of the T2/FLAIR signal with some internal areas showing increased signal intensity and some fluid levels, and peripheral “ring-enhancement” post-contrast, compatible with brain abscesses. The largest abscesses were in the frontal (4 × 3.7 cm) and parietal (5.5 × 7.3 cm) lobes. Additionally, there was evident edematous swelling of the right hemisphere brain tissue, causing a midline shift of about 1 cm. An echocardiogram, a chest X-ray, and an abdominal ultrasound were performed and ruled out the presence of additional infectious foci. Additionally, blood culture results were negative on three sets. Empirical antibiotic therapy with ceftriaxone, metronidazole, and vancomycin was initiated along with anti-edema therapy using dexamethasone.

For diagnostic and therapeutic purposes, surgical drainage of the parietal abscess collection was performed, extracting 25 mL of frankly purulent material. With the aid of an intraoperative trans-fontanel ultrasound, an intracystic catheter was left in place within the parietal abscess, so that 10–15 mL of pus could be removed by the neurosurgeon approximately every 3 days. Methicillin-sensitive *Staphylococcus aureus* (MSSA) was isolated from the pus; therefore, the anti-staphylococcal therapy was optimized by switching from vancomycin to linezolid with the aim to increase tissue penetration and add activity on toxin production. Linezolid was administered intravenously at a dosage of 10 mg/kg per dose every 8 h, and it was infused continuously over 1 h.

In order to study the intracystic permeability of linezolid, its concentration in the intracystic pus was monitored. Ten days after initiating linezolid therapy, steady-state drug levels were measured immediately before (Cmin) and 30 min after a 1 h intravenous infusion to assess peak plasma concentration (Cmax) alongside plasma levels, by using a high-performance liquid chromatography (UHPLC) system (Figure 2).

After 4 weeks of both systemic therapy and local drainage, an MRI scan showed a reduction in size of the parietal collection, while the frontal collection remained unchanged (4 × 3.7 cm). Therefore, it was decided to perform a second surgical drainage, this time of the frontal collection. Once again, the drained material was frankly purulent and tested positive for MSSA. The drainage was left in place for 3 days, up until drainage exhaustion.

The follow-up brain MRI performed at 8 weeks of therapy (Figure 1b) documented a dimensional reduction in the right frontal collection as well (1.3 × 1.2 cm) and a gliotic malacic evolution of the frontal region, caudate, and genu of the right corpus callosum. After the second drainage and confirmation of MSSA through microbiological testing, therapeutic de-escalation was initiated. Overall, the patient completed a 5-week course of metronidazole, a 6-week course of intravenous linezolid, and a 9-week course of intravenous ceftriaxone. Blood counts and liver and renal function tests were monitored at least once a week. No drug-related side effects have been reported. Dexamethasone was used as a full dose for two weeks and then slowly tapered until complete discontinuation after 5 weeks of therapy, in light of the reduction in the edema.

The infant was discharged in good clinical condition without clinical neurological issues. After 6 weeks from the discontinuation of intravenous antibiotic therapy, an MRI showed an almost complete resolution of the known abscesses (the largest being 8 × 7 mm). At the neurodevelopmental assessment at 9 months, the patient exhibited a mild delay in acquiring motor milestones and no other emerging issues.

## 4. Discussion

Brain abscesses are invasive infections requiring complex therapeutic management. They often necessitate both prolonged intravenous antibiotic therapy and a surgical approach.

The recent guidelines issued by the European Society of Clinical Microbiology and Infectious Diseases (ESCMID) on the diagnosis and treatment of brain abscesses in adults and children [8] recommend initiating empirical therapy with third-generation cephalosporin combined with metronidazole in cases where the pathogen is suspected to be community-acquired. Intravenous therapy for at least 6–8 weeks is recommended even if the abscess has been drained. A short therapy of 4 weeks is considered only if surgical excision of the abscess has been possible. There is insufficient evidence to switch to oral antibiotic therapy.

The addition of vancomycin to the first-line therapy is generally suggested when *S. aureus* infection is suspected [1], i.e., penetrating trauma, previous neurosurgery, and hematogenous spread from endocarditis. However, in Italy, according to the National Surveillance System of Antimicrobial Resistance (AR-ISS), the rate of MRSA responsible for invasive infections is around 30%, supporting the use of anti-MRSA therapy when waiting for pathogen identification [9].

In the case described here, the addition of vancomycin to the first-line therapy was also supported by the previous history of admission to the Neonatal Intensive Care Unit (NICU) and *S. aureus* bacteremia treated with teicoplanin.

After MSSA identification in the purulent material, anti-staphylococcal therapy was optimized by switching to linezolid in addition to the ongoing therapy with ceftriaxone and metronidazole. Several studies have described a better tissue permeability of linezolid, including across the blood–brain barrier, compared to vancomycin [6,7]. Additionally, linezolid inhibits the initiation phase of bacterial protein synthesis with effects on toxin production [10]. However, to date, limited literature evidence is available on the brain abscess penetration of linezolid.

Luque S et al. (2014) published an observational PK study involving n = 11 critically ill patients with proven or suspected CNS infection treated with linezolid. The authors reported that the median area under the curve (AUC) value was 47.6 mg*h/L in plasma and 21.1 mg*h/L in CSF, with a median CSF/plasma ratio of 0.77. Moreover, in the steady state, a positive correlation was observed between linezolid concentrations in CSF and plasma, demonstrating that linezolid has a good penetration into the CSF [11].

However, a recent study aiming to predict linezolid cranial CSF profiles based on measured plasma concentrations in adult and pediatric patients affected by tubercolous meningitis showed that, differently from adults, 44% of the pediatric patients did not reach the PK/PD threshold in cranial CSF [12].

The pharmacokinetic/pharmacodynamic (PK/PD) target of efficacy for the time-dependent antibacterial activity of linezolid against methicillin-resistant (MR) staphylococci and against vancomycin-resistant (VR) enterococci has been defined as plasma C_min_ ≥ 2 μg/mL and/or AUC/MIC ratios of >80 [13] (Pea F. et al., 2010). The rationale behind these targets is based on the finding that MIC_90_ for linezolid against both MR staphylococci and VR enterococci is 2 μg/mL [14], and that in one study conducted on seriously ill adult patients, higher success rates were obtained when the drug concentration under steady-state conditions exceeded the MIC for 85% of the dosing interval (%T_MIC_) and the AUC/MIC ratios were between 80 and 120 [15]. Therefore, the achievement of these targets may become ideal in critically ill septic patients [13,15].

In our case, linezolid C_min_ levels in plasma and pus were comparable (1.64 vs. 1.63 μg/mL) and were close to the TDM desired target for C_min_ (2–8 mg/L) [16]. However, 30 min after a 1 h intravenous infusion, the penetration of linezolid in the site of infection (pus) reached a C_max_ of 3.15 μg/mL (higher than 2 μg/mL); meanwhile, linezolid plasma levels at the same time point were 13.52 μg/mL (Figure 2). These data are novel in the literature and confirm the ability of linezolid to penetrate a brain abscess. However, further studies are needed to assess whether higher doses of linezolid could enhance the brain tissue penetration of the drug while maintaining an adequate safety profile and avoiding the need for surgery. Nowadays, a TDM-based approach is necessary to verify the achievement of a PK/PD target, and to ensure optimal therapeutic efficacy.

However, because of the dramatic extent and severity of the infectious process in our patient, a two-stage surgical intervention associated with repeated pus aspiration via an intracystic catheter, in addition to antibiotic therapy, was necessary to achieve a reduction in the volume of the lesions. This is in line with current guidelines that strongly recommend neurosurgical aspiration or excision whenever possible [8] for both diagnostic and therapeutic purposes.

## 5. Conclusions

In summary, the management of brain abscesses requires a personalized approach incorporating targeted antibiotic therapy, TDM, and timely surgical intervention. This case confirms that linezolid is an optimal drug in the treatment of complicated Gram-positive MSSA CNS infections, although an integrated, multidisciplinary treatment strategy is required for a successful outcome.

## Figures and Tables

**Figure 1 pathogens-14-00004-f001:**
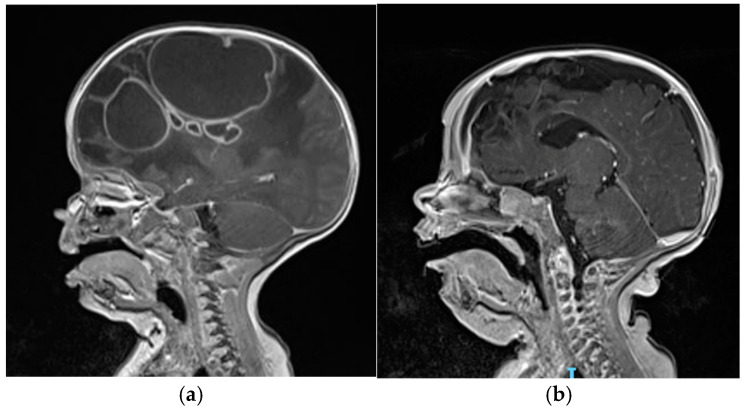
(**a**) The brain MRI scan at the admission showing the presence of multiple brain abscesses in the right hemisphere, the largest in the frontal (4 × 3.7 cm) and parietal (5.5 × 7.3 cm) lobes; (**b**) brain MRI performed after 8 weeks of intravenous antibiotic therapy and two surgical drainages revealing significant reduction in the lesions and a gliotic malacic evolution of parenchyma.

**Figure 2 pathogens-14-00004-f002:**
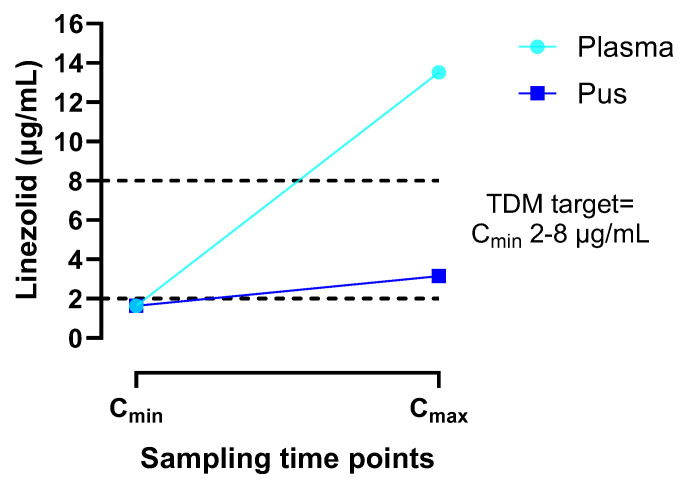
Linezolid concentrations in both plasma (light blue) and intracystic pus (dark blue) samples were measured immediately before (C_min_) and 30 min after a 1 h intravenous infusion (C_max_), ten days after initiating linezolid therapy. Dashed black lines indicate the desired concentration range for linezolid TDM.

## Data Availability

The original contributions presented in the study are included in the article, further inquiries can be directed to the corresponding author.

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
