# Peer review of "Therapeutic Drug Monitoring-Guided Linezolid Therapy for the Treatment of Multiple Staphylococcal Brain Abscesses in a 3-Month-Old Infant"

_pathogens, 2024, doi:10.3390/pathogens14010004_

Round 1
Reviewer 1 Report
Comments and Suggestions for Authors
A 3-month-old infant experienced brain abscess infected by MSSA. The brain abscess was treated by intravenous injection of linezolid and 2-time surgical drainages. At the time of surgical drainage, the authors measured steady-state drug levels (Cmin and Cmax), and confirmed concentrations of linezolid in the pus were preserved above Cmin. Therefore, TDM-based antibiotic therapy in pus could be important for treatment for brain abscess.
The manuscript is well written. However, several minor comments were raised.
#1. In line 50, ‘as important favourable conditions’ may be wrong? I suppose that ‘as important unfavourable conditions’ may be correct.
#2. In line 124, Fig. 3 must be replaced with Fig.2.
#3. In lines 176-177, Is ‘an activity on bacterial toxin production [10]’ decreased? This part is not a sentence, so the meaning is unclear.
#4. “Informed Consent Statement: Informed consent was obtained from all subjects involved in the study.” should be corrected. “Informed consent was obtained from the patents of the patient in the case report.” Is correct.
Author Response
Comment #1. In line 50, ‘as important favourable conditions’ may be wrong? I suppose that ‘as important unfavourable conditions’ may be correct.
Response #1: We modified the sentence to clarify the point.
Comment #2. In line 124, Fig. 3 must be replaced with Fig.2.
Response #2: We have corrected the number.
Comment #3. In lines 176-177, Is ‘an activity on bacterial toxin production [10]’ decreased? This part is not a sentence, so the meaning is unclear.
Response #3: We agree. We modified the sentence in line 180 as follows: “Additionally, linezolid inhibits the initiation phase of bacterial protein synthesis with effects on toxin production”.
Comment #4. “Informed Consent Statement: Informed consent was obtained from all subjects involved in the study.” should be corrected. “Informed consent was obtained from the patents of the patient in the case report.” Is correct.
Response 4#: We agree. We modified the sentence as suggested at line 234.
Reviewer 2 Report
Comments and Suggestions for Authors
Dear Authors,
I express my sincere gratitude for the opportunity to review your manuscript. The effort of the authors is appreciated.
Upon reviewing your manuscript, several issues have raised my concerns:
- Abstract- An MSSA was identified. What was the rationale for switching from vancomycin to linezolid instead of to an anti-staphylococcal agent that is effective against MSSA and has good cerebrospinal fluid penetration?
- “This can be readily explained by considering that the intracranial spread of these pathogens occurs in the majority of cases (50%)” - It is half, not the majority of cases
- “Line 91-93: On this occasion, blood tests were normal, but a blood culture resulted positive for S. aureus (susceptibility profile not known). He received intramuscular therapy with teicoplanin for 3 days, then discharged after clinical improvement with instructions to continue oral amoxicillin-clavulanate for 7 days” - The identification of a positive blood culture with Staphylococcus aureus requires prolonged antibiotic therapy – I don’t understand why this was not performed, especially since teicoplanin was still effective for 24 hours. I also don’t understand the administration of amoxicillin-clavulanate at home, as its efficacy against Staphylococcus is unlikely.
- “Line 99: Blood tests revealed leucocytosis and elevated inflammatory markers (C-reactive protein 3.8 mg/dl).” Provide reference values
- “Line 115-117: Methicillin-sensitive Staphylococcus aureus (MSSA) was isolated from the pus, therefore the anti-staphylococcal therapy was optimized by switching from vancomycin to linezolid with the aim to increase tissue penetration and add activity on toxin production”- A bactericidal anti-staphylococcal agent with good cerebral penetration was replaced by a bacteriostatic one. How do you comment on this? Additionally, the usual administration is every 12 hours, but it was administered at 12 hours?
- Treatment with linezolid for more than 2 weeks can be associated with multiple side effects, especially hematological ones. Was the infant monitored from this perspective?
- “Line 134- After the second drainage and confirmation of MSSA through microbiological testing, therapeutic de-escalation was initiated” - MSSA was isolated well in advance, according to the presentation, and de-escalation should have been performed from the beginning. In the presence of a known etiology, I don't see the reason for continuing treatment with metronidazole, nor the combination of ceftriaxone with linezolid. Additionally, dexamethasone was administered for 5 weeks, supposedly for its anti-edema effect, for which a much shorter period of use would have been sufficient.
In discussions, literature data about the use of linezolid for MRSA are presented, which are inadequate for the strain isolated in this case.
There are two MSSA isolates in the same section, from which we do not obtain an antibiogram for this case. The strain was certainly sensitive to ceftriaxone or cotrimoxazole, rifampicin, or clindamycin, antibiotics that could have been safely used at this age. The linezolid-ceftriaxone combination, bacteriostatic-bactericidal, is not the best alternative in these cases, especially since their action is not synergistic, and the use of metronidazole, given we have an etiology, is neither useful nor beneficial. The case, from a therapeutic point of view, unfortunately evolved unfavorably from the beginning due to a lack of attention to a positive blood culture with MSSA. The manuscript itself presents a bold therapy with linezolid in a pediatric patient, which is very important, but the case itself was not correctly managed.
Author Response
Comments 1: Abstract- An MSSA was identified. What was the rationale for switching from vancomycin to linezolid instead of to an anti-staphylococcal agent that is effective against MSSA and has good cerebrospinal fluid penetration?
Response 1: Although the 2024 ESCMID guidelines on the management of brain abscesses do not specifically recommend a combined antibiotic regimen, we opted to maintain a combination of a beta-lactam and linezolid based on several patient-specific factors. These included the patient's very young age, suspicion of an underlying primary immunodeficiency, and the extensive nature of the brain lesions. Linezolid was chosen over vancomycin primarily due to its pharmacokinetic advantages and safety profile. While no randomized clinical trials directly compare linezolid and vancomycin in this setting, the ESCMID guidelines state: “Vancomycin, a bactericidal glycopeptide, has been the preferred treatment of brain abscess and other central nervous system infections caused by methicillin-resistant S. aureus. However, linezolid has more favourable pharmaco-kinetic properties and experience with this bacteriostatic drug for brain abscess is increasing”. Given the need for prolonged therapy, linezolid was favored to minimize the risk of nephrotoxicity, a known concern with vancomycin, while also ensuring optimal penetration into brain tissue.
Comments 2 “This can be readily explained by considering that the intracranial spread of these pathogens occurs in the majority of cases (50%)” - It is half, not the majority of cases
Response 2: Thanks for your comment. We modified the sentence as follows “spread of these pathogens occurs in the half of cases (50%) through contiguity” at line 42-43.
Comments 3 “Line 91-93: On this occasion, blood tests were normal, but a blood culture resulted positive for S. aureus (susceptibility profile not known). He received intramuscular therapy with teicoplanin for 3 days, then discharged after clinical improvement with instructions to continue oral amoxicillin-clavulanate for 7 days” - The identification of a positive blood culture with Staphylococcus aureus requires prolonged antibiotic therapy – I don’t understand why this was not performed, especially since teicoplanin was still effective for 24 hours. I also don’t understand the administration of amoxicillin-clavulanate at home, as its efficacy against Staphylococcus is unlikely.
Response 3: We agree with this comment. However, it is important to clarify that the patient's initial admission and treatment occurred at a different hospital, and we do not have access to detailed clinical or microbiological data to fully explain the rationale behind the antibiotic regimen chosen by our colleagues at that time. If the current description seems unclear or raises concerns, we can certainly revise it for clarity. One option could be to simplify the sentence as follows: "At 16 days of life, the patient was admitted due to Staphylococcus aureus bacteremia and cellulitis of the right hand, at the site of a previous peripheral venous access, and was treated for ten days." This revised phrasing would focus on the relevant clinical episode without including details about the prior antibiotic regimen, which we cannot fully verify or explain.
Comments 4 “Line 99: Blood tests revealed leucocytosis and elevated inflammatory markers (C-reactive protein 3.8 mg/dl).” Provide reference values
Response 4: Reference values were provided.
Comments 5 “Line 115-117: Methicillin-sensitive Staphylococcus aureus (MSSA) was isolated from the pus, therefore the anti-staphylococcal therapy was optimized by switching from vancomycin to linezolid with the aim to increase tissue penetration and add activity on toxin production”- A bactericidal anti-staphylococcal agent with good cerebral penetration was replaced by a bacteriostatic one. How do you comment on this? Additionally, the usual administration is every 12 hours, but it was administered at 12 hours?
Response 5:
Thank you for your insightful comment. At the time of the switch from vancomycin to linezolid, the patient did not have a septic appearance, and blood cultures were negative. This clinical stability allowed us to consider the use of a bacteriostatic agent like linezolid, particularly given its superior tissue penetration and its ability to inhibit toxin production, which was a relevant concern in this case.
Regarding the dosing of linezolid, the recommended regimen for infants and children under 12 years of age is indeed 10 mg/kg every 8 hours.
Comments 6: Treatment with linezolid for more than 2 weeks can be associated with multiple side effects, especially hematological ones. Was the infant monitored from this perspective?
Response 6: Yes, the patient was monitored with blood counts and liver and renal function tests at least once a week and reported no drug-related side effects. We added a sentence to clarify this point (line 140).
Comments 7 “Line 134- After the second drainage and confirmation of MSSA through microbiological testing, therapeutic de-escalation was initiated” - MSSA was isolated well in advance, according to the presentation, and de-escalation should have been performed from the beginning. In the presence of a known etiology, I don't see the reason for continuing treatment with metronidazole, nor the combination of ceftriaxone with linezolid. Additionally, dexamethasone was administered for 5 weeks, supposedly for its anti-edema effect, for which a much shorter period of use would have been sufficient.
Response 7: Thank you for your thoughtful comments. We opted to continue metronidazole despite the isolation of MSSA, because of the well-documented challenges in detecting anaerobic organisms in cultures. This decision was made to ensure broad coverage, particularly in the context of a brain abscess, where anaerobic involvement cannot be entirely ruled out based on culture results alone. As for the combination of ceftriaxone and linezolid, we have already addressed the rationale for this choice in a previous response, which was based on the patient's young age, pharmacokinetic considerations, and the extent of the brain lesions. Regarding dexamethasone, it was initially administered at a full dose for two weeks to manage cerebral edema, and then gradually tapered over a total period of five weeks. We have added a clarifying sentence (line 142) to the text to explain the tapering schedule and the rationale behind its extended use, which was tailored to the patient's response and clinical needs.
Comments 8: In discussions, literature data about the use of linezolid for MRSA are presented, which are inadequate for the strain isolated in this case.
Response 8: Thank you for your observation. We acknowledge that the literature on the use of linezolid for MSSA infections is limited, primarily because other antibiotics with excellent activity against MSSA are more commonly used, and linezolid is often reserved for more complex cases due to its higher cost and specific indications. However, in the context of brain abscesses—rare but serious infections—pharmacokinetic and pharmacodynamic (PK/PD) data on linezolid, particularly in young children, remain scarce. Given the challenges of collecting robust data on antibiotic brain penetration in clinical settings, especially in paediatric patients, our decision was based on the available evidence and clinical judgement. While most studies focus on MRSA, we believe that sharing our experience with the use of linezolid in cases of both MSSA and MRSA in brain abscesses can contribute to expanding the knowledge base, potentially aiding other centres in managing these severe infections.
Comments 9:
There are two MSSA isolates in the same section, from which we do not obtain an antibiogram for this case. The strain was certainly sensitive to ceftriaxone or cotrimoxazole, rifampicin, or clindamycin, antibiotics that could have been safely used at this age. The linezolid-ceftriaxone combination, bacteriostatic-bactericidal, is not the best alternative in these cases, especially since their action is not synergistic, and the use of metronidazole, given we have an etiology, is neither useful nor beneficial. The case, from a therapeutic point of view, unfortunately evolved unfavorably from the beginning due to a lack of attention to a positive blood culture with MSSA. The manuscript itself presents a bold therapy with linezolid in a pediatric patient, which is very important, but the case itself was not correctly managed.
Response 9: In our previous responses, we explained the rationale behind our antibiotic choices. We would like to clarify that, during the patient’s stay at our hospital, no positive blood cultures were recorded. The patient had a positive blood culture during a prior admission at another hospital, but we do not have information regarding the antibiotic susceptibility of that strain.
There could be a discuss about whether the patient was overtreated for brain abscesses. However we would emphasize the complexity of assessing clinical response to therapy in an infant who is afebrile, has insignificant inflammatory markers, and whose neurological status is challenging to evaluate due to his age. The purpose of our case report was not to provide guidance on the targeted therapy for MSSA brain abscesses but rather to share information on the brain penetration of linezolid in a very young child. These data may assist other clinicians for managing paediatric cases of serious CNS infections in the future.
Reviewer 3 Report
Comments and Suggestions for Authors
Manuscript ID: Pathogens-3205600
Title: Linezolid brain penetration: a TDM-based approach for the treatment of multiple staphylococcal brain abscesses in a 3-month-old infant
Review Summary:
This is a case report describing experience of treatment of an infant with multiple brain abscesses consisting of a combined surgical (abscess drainage) and antibiotic (linezolid) approach. Therapeutic Drug Monitoring (TDM) was used to evaluate linezolid concentrations in both plasma and abscess drainage fluid to support comparison to recommended therapeutic levels based on plasma concentrations. Standard regimen for linezolid treatment in pediatric patients was used.
The paper is easy to follow. This reviewer has a few comments (below) to improve its impact.
Major comments:
It is not clear on what day of linezolid therapy the specimens of plasma and pus were taken for measurement of linezolid concentrations. The authors state that the levels were taken at steady-state; still, it would be useful to indicate the specific day or indicate after the specific dose following initiation of linezolid therapy.
Minor comments:
Figure 2: It would be helpful to add in the figure caption which day after initiating linezolid therapy the measures were taken.
The authors should mention that linezolid therapy alone is apparently insufficient to treat the brain abscesses, since a second drainage (surgical intervention at a different site) was needed 4 weeks after initiating linezolid therapy. There is discussion that linezolid, based on some publications, can penetrate the CNS; however, the regimen applied by the authors does not support adequate penetration to resolve an infection based on linezolid therapy alone. The authors should discuss that additional studies would be needed to understand if higher linezolid doses could be used to treat CNS infections as monotherapy (i.e., not requiring combined surgical intervention).
Conclusion: The authors may want to add “MSSA” after “Gram-positive” in the sentence, “This case confirms that linezolid is an optimal drug in the treatment of complicated Gram-positive CNS infections,….”
Author Response
Comments 1 e 2:
Major comments:
It is not clear on what day of linezolid therapy the specimens of plasma and pus were taken for measurement of linezolid concentrations. The authors state that the levels were taken at steady-state; still, it would be useful to indicate the specific day or indicate after the specific dose following initiation of linezolid therapy.
Figure 2: It would be helpful to add in the figure caption which day after initiating linezolid therapy the measures were taken.
Response:
Thank you for your comment. We added the information at line 124 and 156.
Comments 3
The authors should mention that linezolid therapy alone is apparently insufficient to treat the brain abscesses, since a second drainage (surgical intervention at a different site) was needed 4 weeks after initiating linezolid therapy. There is discussion that linezolid, based on some publications, can penetrate the CNS; however, the regimen applied by the authors does not support adequate penetration to resolve an infection based on linezolid therapy alone. The authors should discuss that additional studies would be needed to understand if higher linezolid doses could be used to treat CNS infections as monotherapy (i.e., not requiring combined surgical intervention).
Response 3: Thank you for the comment. We modified the sentence as follows: “These data are novel in the literature and confirm that linezolid can penetrate the brain abscess. However, it is important to note that in our case, linezolid monotherapy was not sufficient to fully resolve the infection, as a second surgical drainage was necessary. Further studies are needed to evaluate whether higher doses of linezolid could enhance penetration into brain tissue, maintaining an adequate safety profile while potentially avoiding the need for surgery. A TDM-based approach is currently essential to verify whether PK/PD targets are achieved, ensuring optimal therapeutic efficacy”.
Comments 4
Conclusion: The authors may want to add “MSSA” after “Gram-positive” in the sentence, “This case confirms that linezolid is an optimal drug in the treatment of complicated Gram-positive CNS infections,….”
Response 4: We modified the sentence.
Round 2
Reviewer 2 Report
Comments and Suggestions for Authors
The authors provide detailed answers to my comments. The quality of the case report improved. The manuscript from my point of view is suitable for publishing in this form.
Author Response
Thank you for the interest in the theme and for the suggested changes.